# X-Wines: A Wine Dataset for Recommender Systems and Machine Learning

Rogério Xavier de Azambuja [1,2,3,*], A. Jorge Morais [2,4] and Vítor Filipe [3,5]

1 Instituto Federal do Rio Grande do Sul (IFRS), Farroupilha 95174-274, RS, Brazil
2 Department of Science and Technology, Universidade Aberta (UAb), 1269-001 Lisbon, Portugal
3 School of Science and Technology, Universidade de Trás-os-Montes e Alto Douro (UTAD), 5000-801 Vila Real, Portugal
4 LIAAD—INESC TEC, 4200-465 Porto, Portugal
5 INESC TEC—INESC Tecnologia e Ciência, 4200-465 Porto, Portugal
* Correspondence: rogerio.xavier@farroupilha.ifrs.edu.br

**Abstract:** In the current technological scenario of artificial intelligence growth, especially using machine learning, large datasets are necessary. Recommender systems appear with increasing frequency with different techniques for information filtering. Few large wine datasets are available for use with wine recommender systems. This work presents X-Wines, a new and consistent wine dataset containing 100,000 instances and 21 million real evaluations carried out by users. Data were collected on the open Web in 2022 and pre-processed for wider free use. They refer to the scale 1–5 ratings carried out over a period of 10 years (2012–2021) for wines produced in 62 different countries. A demonstration of some applications using X-Wines in the scope of recommender systems with deep learning algorithms is also presented.

**Keywords:** big data; wine reviews; recommender systems; machine learning; deep learning

## 1. Introduction

Wine is undoubtedly a fascinating product and has a production chain that requires knowledge, talent and creativity. There are those who consider it to be an art product. According to the International Organization of Vine and Wine (OIV) [1], "Wine is exclusively the beverage resulting from the complete or partial alcoholic fermentation of fresh grapes, whether crushed or not, and from the grape must". It acquires its own characteristics-linked structure, aroma and taste, being an alcoholic product widely enjoyed by adults.

The Web offers a flood of information which needs to be properly treated and validated to be useful. Without proper verification and validation, data volumes may be nothing more than simple simulations and may not reflect reality. Similarly, data science through its numerous techniques and aggregated processes inevitably requires initial data, often raw, to be pre-processed and turned into valuable information. The pre-processing work can be very difficult. Filling a detected gap is the main motivation for building a wine dataset and making it openly available for the scientific community.

Currently, there are large reference datasets on movie ratings from GroupLens [2], containing real user ratings in different sizes (100,000, 1 million, 10 million, 20 million and 25 million); electronic items on e-commerce from Taobao [3]; fashion products from Amazon [4]; Iris flower features from R. A. Fisher [5]; books and ratings from Book-Crossing [6] (with anonymized users, but with demographic information); and anonymous data ratings of online jokes from Jester [7] (a dense dataset with a large number of users but small number of jokes). These and other datasets may be found in scientific publications, and there are very few references to the wine domain. Some of them may be found in repositories such as Kaggle Datasets [8] and GitHub Data Packaged Core Datasets [9];

however, they present scarcity of relevant data or are organized without the necessary rigor for a wider use.

Usually, in a similar way to most agricultural products, wines present a very small data volume or with few elements that end up conditioning the work, limiting scientific exploration, as is the case for recommender systems, which transform product instances and historical user data into a personalized and promising product offering through information filtering techniques [10,11]. Due to their utility, recommender systems have gained attention in e-commerce, e-business, e-learning, e-tourism, etc., attracting the interest of companies such as Netflix, Amazon, Google and Alibaba. Customer ratings are considered to provide recommendations, such as MovieLens information filtering algorithms [12], the Amazon portal [13] and Netflix streaming video [14]. These have become well-known datasets frequently quoted and updated in the recommender systems in the literature [15–17]. Research on recommender systems and applications is increasing in e-commerce [18,19], the open social network domain [20,21], multimedia [22,23], healthy behavior [24,25], e-tourism [26,27] and others. In the agricultural field, broad domain work is found, such as crop cultivation suggestion based on a input soil dataset [28] and assisting farmers' inquiries through a collaborative recommender system [29]. Studies in the specific domain such as fertilizer recommendation to farmers [30] and pest and treatment recommendation [31] are also found, with very few involving the wine domain.

To overcome this problem, we present X-Wines, a world wine dataset with user ratings for recommendation systems, machine learning and general purpose, which is a new open dataset to be freely used in research and educational projects. A simple demonstration of the applications using X-Wines in classic recommender systems is presented in this paper to recommend relevant items to users measured by evaluation metrics. In Section 2, the work methodology is presented, including data collection, dataset description, verification and validation with benchmark analysis. In Section 3, a demonstration of its use and results measurement is shown. Finally, the paper ends with conclusions and possible future work.

## 2. Work Methodology

Ethical and scientific principles were followed to build a world wine dataset for wider use. The X-Wines dataset construction was carried out over a period of six months in 2022, organized in two stages: data collection and its posterior verification and validation.

### 2.1. Data Collection Process

The data collection process was carried out on several wine-specialized websites on the open Web between the months of February and March in 2022. The Python programming language version 3.9 (Python Software Foundation, Wilmington, DE, USA) was used to collect and process the metadata. Firstly, an exploratory search was carried out for wine items candidates and wineries' websites. Basically, the search algorithms tried to detect keywords in metadata, assigned scores and stacked candidate URLs and URIs. These sites make available on the Web attributes identified as wine objects and their real instances. The ratings carried out by real users registered on the platforms followed the processing protocol after the user's authorization via the website to publish their evaluations and thus help other people in an open social networking format.

It is important to highlight that no systems were security-checked or breached. Data collection was carried out only on public data available on the open Web. Our results, detailed below, do not use private data, respecting data protection and privacy laws. Portals were officially contacted and informed about this research, and there were no objections that prevented the continuation of the data collection. Some wineries were grateful for the exposure and dissemination of their products. Main reference sources were explored; among them, more than 350 were considered useful for providing systematic data. Several identified references required specific adaptations in the exploratory algorithms, and in some cases the algorithms were reconstructed with their own functionalities adapted to specific sources. The used sources were made available in the official repository of the

X-Wines project at the address https://github.com/rogerioxavier/X-Wines (accessed on 26 December 2022).

The data collected estimated the existence of 228,000 wine items that could be used. Texts and unmarked images that could build different attributes to one wine instance were collected. They were identified and stored, and after they were pre-processed and validated in the process described in Section 2.3. Numerous candidate attributes were found in various types and formats. Figure 1 presents some of the selected attributes: elaborate, grapes, harmonize, alcohol by volume (ABV), body and acidity. The criteria for selecting the attributes considered the frequency and the volume of these attributes available on specialized websites and also their presented formats, in consideration of the future possibility for electronic verification and validation by specific algorithms.

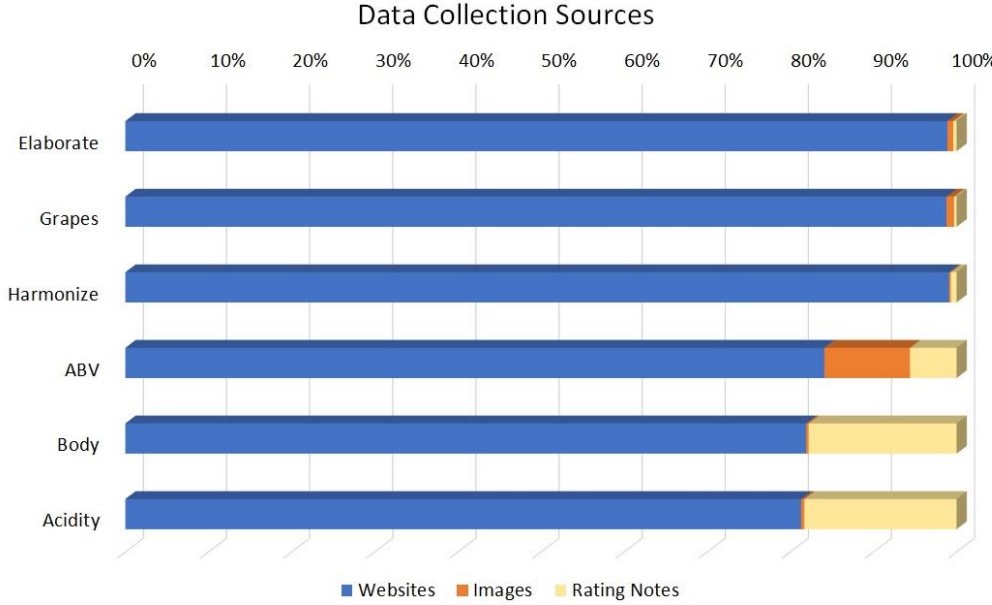

**Figure 1.** Distribution of attributes collected by source types.

Most data were collected in textual format in *html*, *json* and *xml* formats or from files in *pdf*, *txt* and *epub* formats, being complemented with data obtained by extracting information from images on the respective wine labels. As illustrated in Figure 1, most attributes were taken directly from websites, while some attributes were found in abundance on wine label images available on the Web and in rating notes produced by the users. When information was not found explicitly on the websites, including the official producers' websites, which sometimes did not provide important information about their products, specific text extraction, natural language processing and selection algorithms were applied in these cases to find relevance in data.

Optical Character Recognition (OCR) was applied to explore the label image shear on the wine bottle. It is often curved and in several different positions, factors that made it difficult to standardize the data extraction from images. With the additional experimentation of several image filters presented in Figure 2, we tried to calibrate the textual recognition algorithms from the labels.

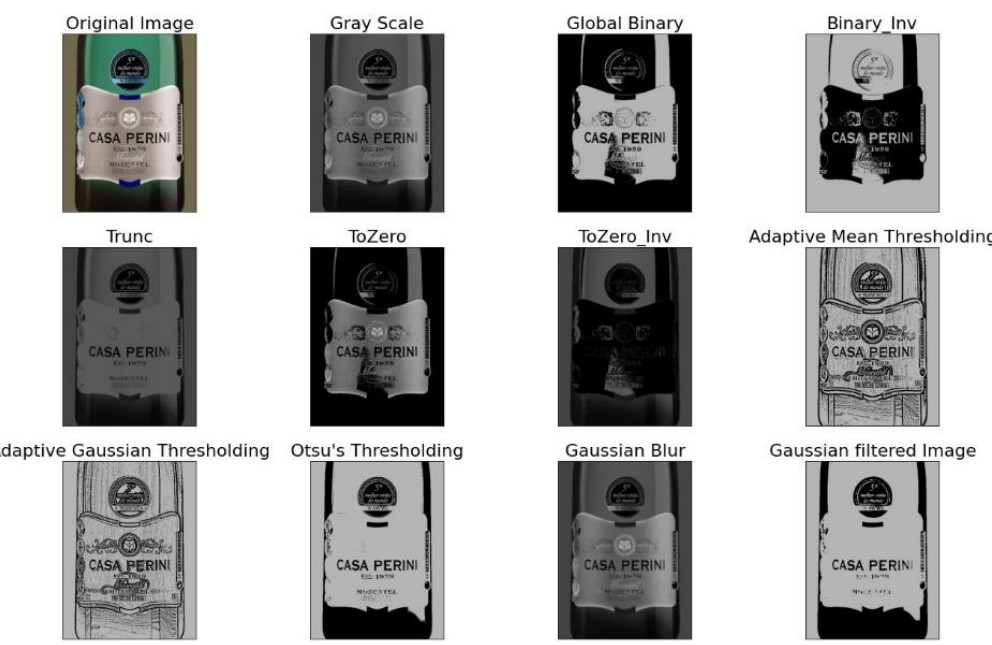

**Figure 2.** Image calibration pre-processing for OCR algorithms.

Image filters were toggled and combined to provide inputs, and *pytesseract* library [32] algorithms were used for the OCR process. The best results were obtained in the text extraction from the *trunc* filter (*THRESH_TRUNC* from the OpenCV library [33]) in which the pixel values above the threshold were set to the same as the threshold, and all other values remained the same. In combination, adaptive Gaussian thresholding and even using the simpler Gaussian blur method brought good results in some items transformed into grayscale from the original image. Figure 1 illustrates in orange color the 10,278 ABV values found in the respective label images. Certainly, this performance can be improved in the future with the application of new improved algorithms which can support the data extraction of wine label images more accurately. It is important to mention that many labels do not present information on alcohol content because laws are different from one country to another. Proper names referring to the wine, winery and region identification were also found in the label images; however, many formed partial words and were used as substrings in the data verification and validation process. The selected unmarked images organized in *jpeg* format and 480 × 640 pixel resolution size accompany the X-Wines dataset in the official repository.

The user comments were anonymously processed without the intention of correction but with the purpose of collecting attributes frequently presented because 63 different languages were detected. Natural language processing was performed by specific algorithms such as the *Translator* package from the *Googletrans* library [34] and the *spacy_langdetect* library [35]. We sought to find out whether a particular wine was expressed as very light, light, medium, full or very full-bodied or else low, medium, medium-, medium+ or high acidity by the majority (greater than 75%, for example) among all users who drank and rated it.

*2.2. Dataset Description*

The built dataset is in line with findable, accessible, interoperable and reusable (FAIR) guiding principles [36], enabling computational resources to automatically find and use the data. In this way, metadata with a unique and persistent identifier were utilized to increase findability. X-Wines is openly available and will be published under a free license in the official repository. However, it is possible to provide better accessibility to individual wines and their attributes through communication protocols and direct links to access points. In addition to some care being taken, a common and applicable language to represent

knowledge in the wine domain was used in order to establish data interoperability, and relevant wine attributes and evaluation sequences by anonymized users were validated with consistency to enhance the data reusability.

The X-Wines dataset is available in three versions: Full, containing all records; Slim, containing one percent of random sample of records and a Test version containing the 100 wines verified and presented below in Section 2.4 and only 1000 ratings for dataset experimentation. Table 1 presents the main comparison between the different versions.

**Table 1.** Characteristics of the presented versions.

| Version | # Wines | # Wine Types | # Wine Countries | # Users | # Ratings | Multiple User Wine Rating |
|---|---|---|---|---|---|---|
| **Test** | 100 | 6 | 17 | 636 | 1000 | No |
| **Slim** | 1007 | 6 | 31 | 10,561 | 150,000 | No |
| **Full** | 100,646 | 6 | 62 | 1,056,079 | 21,013,536 | Yes |

The X-Wines dataset is composed of two files: XWines_100K_wines.csv and XWines_21 M_ratings.csv. Both are detailed below; they keep the same structure, and only the number of records changes in their different versions.

### 2.2.1. XWines_100K_wines.csv File Description

Among different classifications for wine products found on the open Web, the internationally adopted classification [1,37] was used with the following attributes selected:

1. **WineID**: Integer. The wine primary key identification;
2. **WineName**: String. The textual wine identification presented in the label;
3. **Type**: String. The categorical type classification: Red, white or rosé for still wines, gasified sparkling or dessert for sweeter and fortified wines. Dessert/Port is a subclassification for liqueur dessert wines;
4. **Elaborate**: String. Categorical classification between varietal or assemblage/blend. The most famous blends are also considered, such as Bordeaux red and white blend, Valpolicella blend and Portuguese red and white blend;
5. **Grapes**: String list. It contains the grape varieties used in the wine elaboration. The original names found have been kept;
6. **Harmonize**: String list. It contains the main dishes set that pair with the wine item. These are provided by producers but openly recommended on the internet by sommeliers and even consumers;
7. **ABV**: Float. The alcohol by volume (ABV) percentage. According to [1], the value shown on the label may vary, and a tolerance of 0.5% per 100 volume is allowed, reaching 0.8% for some wines;
8. **Body**: String. The categorical body classification: Very light-bodied, light-bodied, medium-bodied, full-bodied or very full-bodied based on wine viscosity [37];
9. **Acidity**: String. The categorical acidity classification: Low, medium, or high, based on potential hydrogen (pH) score [38];
10. **Code**: String. It contains the categorical international acronym of origin country of the wine production (ISO-3166);
11. **Country**: String. The categorical origin country identification of the wine production (ISO-3166);
12. **RegionID**: Integer. The foreign key of the wine production region;
13. **RegionName**: String. The textual wine region identification. The appellation region name was retained when identified;
14. **WineryID**: Integer. The foreign key of the wine production winery;
15. **WineryName**: String. The textual winery identification;
16. **Website**: String. The winery's URL, when identified;
17. **Vintages**: String list. It contains the list of integers that represent the vintage years or the abbreviation "N.V." referring to "non-vintage".

### 2.2.2. XWines_21M_ratings.csv File Description

1.  **RatingID:** Integer. The rating primary key identification;
2.  **UserID:** Integer. The sequential key without identifying the user's private data;
3.  **WineID:** Integer. The wine foreign key to rated wine identification;
4.  **Vintage:** String. A rated vintage year or the abbreviation "N.V." referring to "non-vintage";
5.  **Rating:** Float. It contains the 5-stars (1–5) rating value $\subset$ {1, 1.5, 2, 2.5, 3, 3.5, 4, 4.5, 5} performed by the user;
6.  **Date:** String. Datetime in the format YYYY-MM-DD hh:mm:ss informing when it was rated by the user. It can be easily converted to other formats.

### 2.3. Data Verification and Validation Process

From the collected data, several verifications and validations were performed through electronic processes to constitute the X-Wines dataset. Extensive pre-processing work was carried out over four months, which produced over 3000 source-code lines and led to millions of unapproved data being discarded. Different formats and classifications were validated to bring the data presented here as close as possible to reality.

Due to the large data volume obtained in the collection process, we sought to limit the users' numbers who evaluated only one or a few wines, as well as wine items that received very few evaluations. A cut-point close to 5, for example, tends to increase the consistency between data for data science and allow the data sequence construction that can be explored in scientific experimentation. These sequences can be particularly useful to recommender systems, for example, in the historical data treatment [12–15]. To ensure their relevance in future use, data were pre-processed based on two quantitative limitations:

a.  Only wine items that presented five or more ratings;
b.  Only ratings carried out by users with five or more wine reviews among the selected wines.

After the minimum quantitative cut, the wine instances and their respective attributes were validated. Additional control columns were created in each record, and different business rules based on specific knowledge in the wine domain were exhaustively executed. Dictionaries with words from the wine domain were also used. This involved intensive research and processing. Only when a record was 100% positive in all control columns was it finally validated as machine actionable [36]. The main reference sources of knowledge on the wine domain used in data validation were Wine Folly [37], The Wine Bible [38], Wine Encyclopedia Lexicon [39] and International Organization of Vine and Wine (OIV) [1].

The validated ratings were carried out by users in the period of 10 years between 2012 and 2021, as few reviews were found before the year 2012. Those were discarded, with only those carried out in the period of highest frequency shown in Figure 3 prevailing.

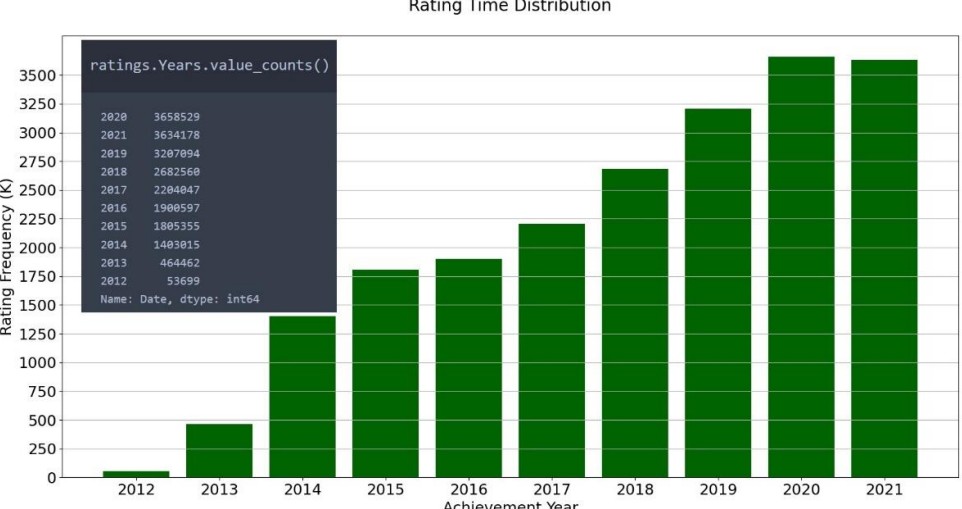

**Figure 3.** Distribution of validated ratings by year in the collected period.

In addition to the validated rating numbers illustrated in Figure 3, involving reviews were performed by users, Figure 4 illustrates the evaluated wine vintages, when the rated wines were produced. This is important because wine is a product with great durability when compared to others. It can be aged for many years in different containers until it is ready for human consumption and has a shelf life of many years [1,38]. The possibility of cataloging the storage time was considered along with numerous other candidate attributes, but this was rarely available in the collected data. The vintages or the grape harvests were identified in abundance.

Regarding the evaluated vintages, only those found in the production range of the last 70 years (approximately, between 1950 and 2021) were considered. Wines that, in addition to the regular annual harvests, use grape varieties harvested in different years of production were also validated. The acronym N.V., "non-vintage", was employed, which is the traditional case of many sparkling wines, for example. Figure 4 shows the vintage distribution per year of the validated sample from the Web. Besides the vintage, other attributes were pre-processed, among them the identification of the valid rated wine, the date on which the reviews were performed and the 5-stars rating value assigned by a valid and anonymized user.

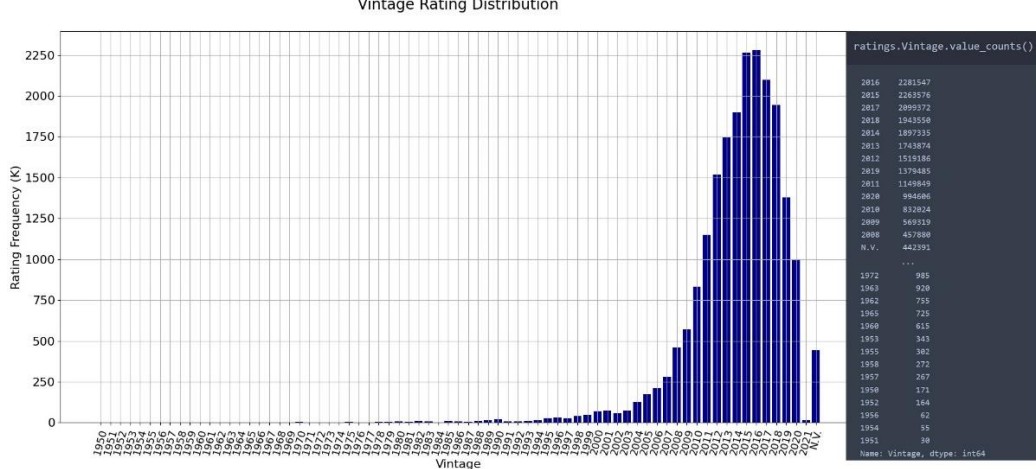

**Figure 4.** Distribution of validated vintages by years.

Validated users were not explicitly identified in X-Wines, preserving their identity, since it becomes more important to record their ratings. The main pre-processing activities performed electronically by classification and consistency algorithms produced by the authors to constitute the X-Wines dataset were:

a.  Removal of duplicate records;
b.  Removal of records with some date inconsistencies in the relationship between wines–ratings–users. Thus, the calendar error in relation to the vintage of that wine was eliminated, for example, exclusion of ratings recorded in 2012 for a specific wine from the 2013 vintage;
c.  Adjusting the wine type to the international classification [1];
d.  Statistical rounding adjustment in numerical values to the 5-stars rating mostly used. For example, 3.9 to 4.0 and 4.6 to 4.5 in the rating column. This was necessary in less than 0.05% of the records found, with only 10,298 adjustments referring to 463 wines;
e.  Optionally, the URLs obtained referring to the winery websites or related to them were kept when their origin was validated, and some data were obtained from them. This was not possible for all wine items found.

When verifying the data origin obtained, winery websites and product datasheets received the highest score in the origin ranking, followed by the data obtained from the specialized wine platforms on the open Web. However, it is important to highlight the fact

that many official websites lack information about these products. Third-party websites, such as e-commerce, were used with lower scores in the origin ranking. Blogs were avoided.

In this way, none of the data presented in the X-Wines dataset were created manually; data were obtained through a collection process on the open Web and validated from different sources. Thus, the following final numbers were obtained:

- 100,646 wine instances containing 17 selected attributes;
- 21,013,536 5-stars rating instances, containing a date and a value in a 1–5 scale;
- 1,056,079 anonymized users.

### 2.4. Statistical Analysis

After carrying out the processes described in previous sections, a preliminary dataset was formed. A statistical test was conducted to estimate confidence of the data obtained from the Web. A simple random sample containing 100 wines was drawn among the validated wines with their respective attributes. The draw was organized in a stratified way considering all six types classified and performed algorithmically in two stages:

a. Five wines were randomly selected for each of the six types found, forming 30% of the sample;
b. Seventy wines were randomly selected, regardless of any attribute, comprising 70% of the sample.

Figure 5 illustrates the characterization of the sample used, which sought to ensure a good variability of the selected wines spread across all types, from different wineries and several countries.

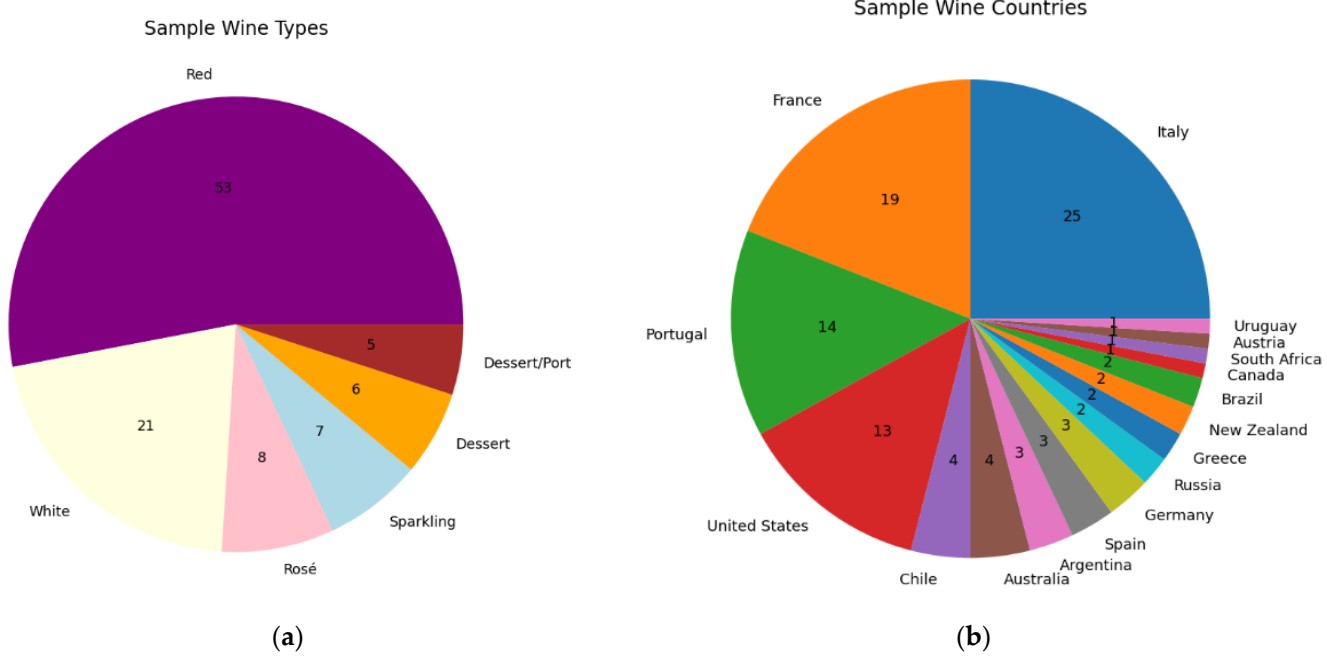

(**a**)                                                              (**b**)

**Figure 5.** Sample distribution by types (**a**) and countries (**b**) in the 100 selected wines.

Ten attributes were individually checked in the document-based benchmark for each wine selected. A total of 1000 pieces of information were checked against data reported by wineries or their representatives. Each one was assigned a score after checking three conditions: value 1 if there was a match, 0.5 it was partially correct with the need for some adjustment or 0 if it did not correspond to the data found. The documents, images and websites used in the document-based benchmark are available in the X-Wines repository. Figure 6 illustrates the analyzed results below.

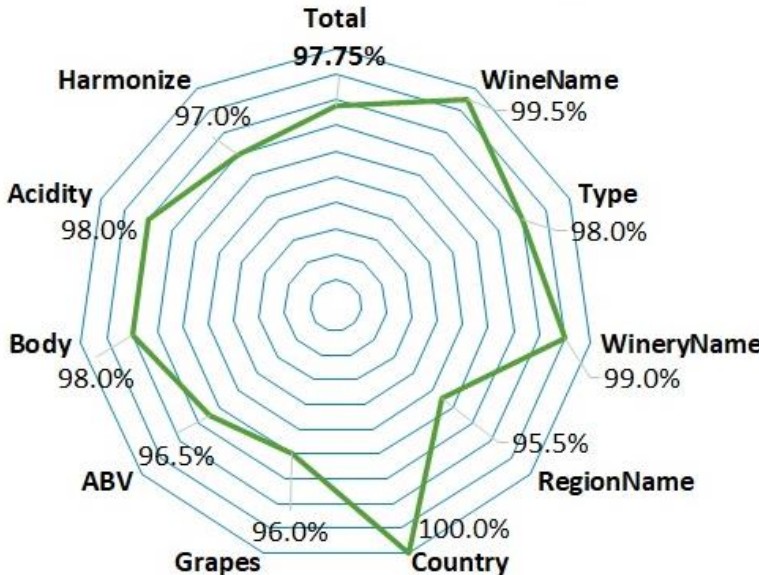

**Figure 6.** Result of the document-based benchmark in the 100 selected wines.

The document-based benchmark global result had a coincidence of 97.75% and standard deviation of 4.15%. This result was obtained by averaging all proportions of data obtained from the Web with the official datasheet from producers. Thus, with a 95% confidence interval, the true mean of assertiveness is between 96.94% and 98.56%. Specifically, the processes described in the previous sections were able to obtain 100% accuracy in the categorical attribute "country". Close results were found when checking the other attributes, with accuracy still high. According to [1,37], in the winemaking process, grape varieties and ABV values may vary by different vintages, explaining some of the discrepancies found in these attributes obtained on the Web. The body and acidity attributes contributed a large amount of information found in the users' evaluations, considering what most of them reported when these data were not found in some sources, and obtained a high accuracy percentage. The harmonize attribute is a very particular issue to be evaluated, and it was considered a match when at least one parity indication coincided. The largest adjustments were necessary in the wine region attribute, containing the name of the region of the wine's origin. The region is usually presented as found on the label (appellation region), but these labels can also display the geographic region or subregion from the producing country or even the region of the winery, which sometimes differs from where the wine was produced. One aspect of the document-based benchmark was the finding two white wine types advertised widely on the Web in this way; however, in the official datasheet they are listed as the dessert wine type, with high levels of residual sugar in the product.

## 3. Classical Application of the Dataset in Machine Learning and Recommender Systems

Aided by hardware evolution (CPUs, GPUs, SSDs, etc.) and communications, machine learning, especially using deep learning, has become the state-of-the-art in areas such as speech and visual object recognition and natural language processing, and it is evolving in the recommender systems area [10,40]. The recommendation problem can be formally defined by finding a utility function to recommend one or more items with the highest estimated ratings, ranked by one output score obtained for each item to the user [41].

There are different approaches to finding a feasible solution to the recommendation problem, mainly collaborative, content-based, knowledge-based and even hybrid. However, several taxonomies are found in the literature marking evolutions in recommender system approaches over time [42]. In [43], the recommendation problem is summarized as a

matrix filling task. Mathematics and statistics add accuracy to human perception, which can be varied to find similar or diverse items. More recently, machine learning methods through deep neural networks (DNNs) have been used [10,44,45]. Some models with DNNs present methods that can switch the calculation between more than one similarity and diversity metric; still, serendipity, novelty and other metrics are explored in the information filtering [46–48]. According to the extensive review conducted in [44], the DNN approach is growing in recent years in recommender systems, mainly using reinforcement learning. Self-attention and bidirectional attention-based models were tested [49] with long-term scenarios represented by dense datasets, simultaneously with sparse datasets generated by more recent user interactions.

Accuracy and the ability to handle large data volumes are advantages of differential machine learning in real-world representations. However, there are open questions regarding the computational performance of DNNs in offline and especially in online environments [50], mainly in predicting the user's short-term interest in a session and the required data for experimentation.

The X-Wines dataset is very wide and allows the construction of multiple data combinations. As a practical demonstration of its usability, two examples built in Python, with wide use of technologies, and a study with metrics commonly used to evaluate recommender systems are presented.

### 3.1. Experimental Applications in Wine Recommendation

Among the various possible approaches to recommender systems found in the literature, the first experimental application performs collaborative information filtering. It predicts wines that a user would prefer based on user neighborhood similarity. The libraries *sklearn.neighbors*, *scipy*, *numpy* and *pandas* were used for processing, as well *matplotlib* and *cv2* for output plotting.

Firstly, a data filter that considers only wines produced in Portugal and their respective ratings was utilized. The major processing on this eventually selected sample was performed in three steps:

1. Pivot table construction for mapping user wine ratings, filling value 0 when there is no such relationship;
2. Sparse matrix treatment where the *csr_matrix* package from the *scipy* library is used;
3. Finally, the identification for each user's neighborhood is processed by *k-nearest neighbor* (KNN) algorithms [51] over the input relationship matrix.

Non-parametric supervised learning was carried out using *brute force*, *BallTree* or *KDTree* (K-Dimensional binary Tree) specific algorithms, being auto by default, to calculate the similarity between the evaluated wines. The default setup of the KNN algorithms was used in training with number of searched neighbors equal to 15 and cosine distance, among the various similarity options found in the module *scipy.spatial.distance*, such as Minkowski, Euclidean, Manhattan and Jaccard.

Two specific functions were created: The function *RecommenderKNN* processes inference in the neighborhood from collaborative user–wine relationship; giving a wine as an input it returns the other N closest instances identified by similarity and the function *show10wines* to graphically display the recommended wine labels from the dataset. As a result, for the first experimental application, a simulation from wine instances is illustrated in Figure 7a, recommending the top-10 Portuguese wines with their similarity measures found. In this example, Portuguese wine "Defesa" was eventually selected as the input.

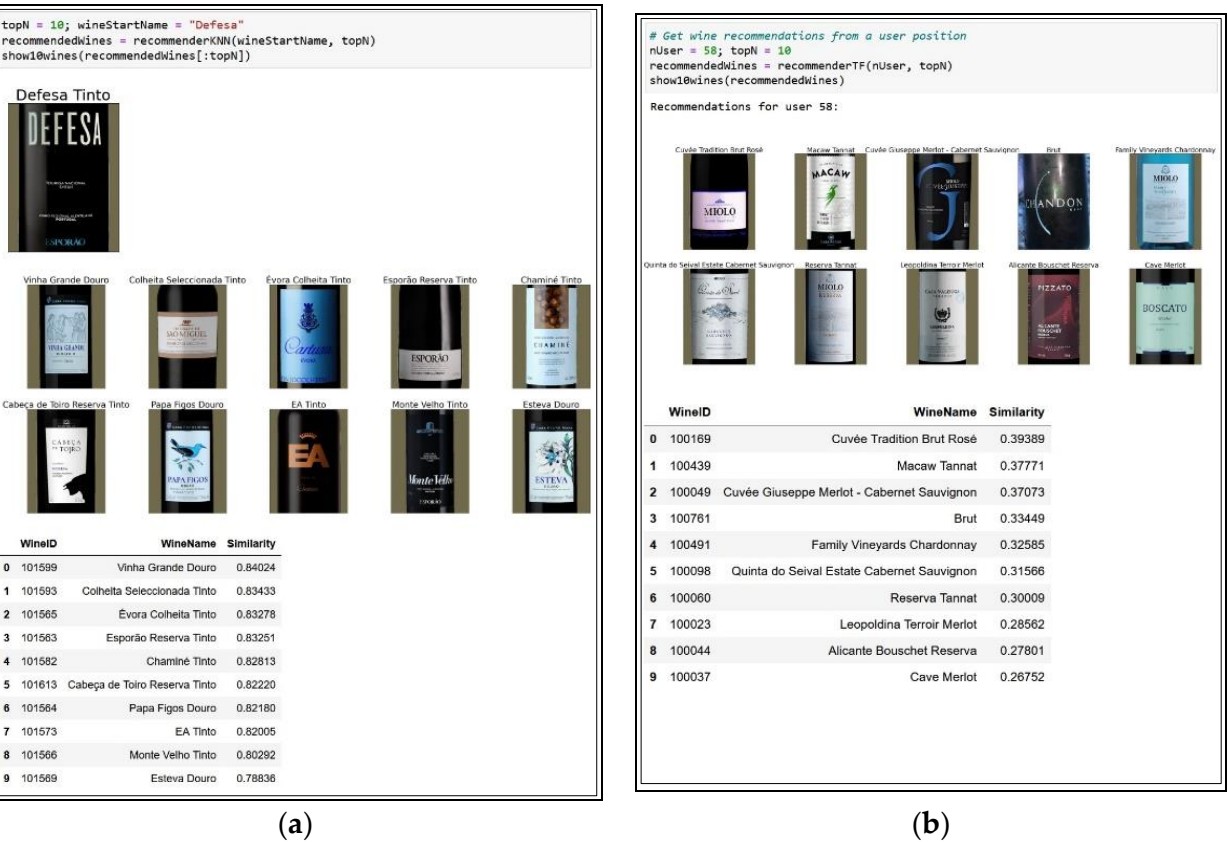

**Figure 7.** Output of the recommender systems using KNN (**a**) and using a basic recommender model with the Tensorflow library (**b**).

Another practical example demonstrates the use of tensors through a basic recommender model with the TensorFlow library [52] for training and testing. The second application demo performs processing to recommend Brazilian wines for an eventually selected user through content-based filtering. The collaboration is considered only between the ratings carried out in Brazilian wines as eventually a data sample. The libraries *pandas*, *numpy*, *typing*, *tensorflow* and specifically *tensorflow_recommenders* were used for processing as well *matplotlib* and *cv2* for image treatment and output plotting.

Firstly, a data filter in the X-Wines dataset to obtain only Brazilian wines and their respective ratings was performed. From this selected sample, two tensors were created: from unique wine identifications and from the unique relationship between user and their rated wines. With them, a machine learning model finds a similarity relationship between wines rated by the same users. The following casual setup was used: random split on shuffled sample between 90% for training and 10% for testing, embedding dimension equal to 32 and learning rate of 0.5 per only 10 epochs.

As a processing result for the second experimental application, the simulation from one user code is presented. Figure 7b illustrates requesting the top-10 Brazilian wines and their similarity measures for a user. In this example, a user code was eventually selected to demonstrate the result. From the model definition used for training during some epochs, the accuracy evaluation by metrics commonly used in recommendation systems, top-N | N $\subset$ {1, 5, 10, 50, 100}, is shown in Figure 8a (named as top_k in Tensorflow). Another traditional deep learning metric is the loss over training shown in Figure 8b.

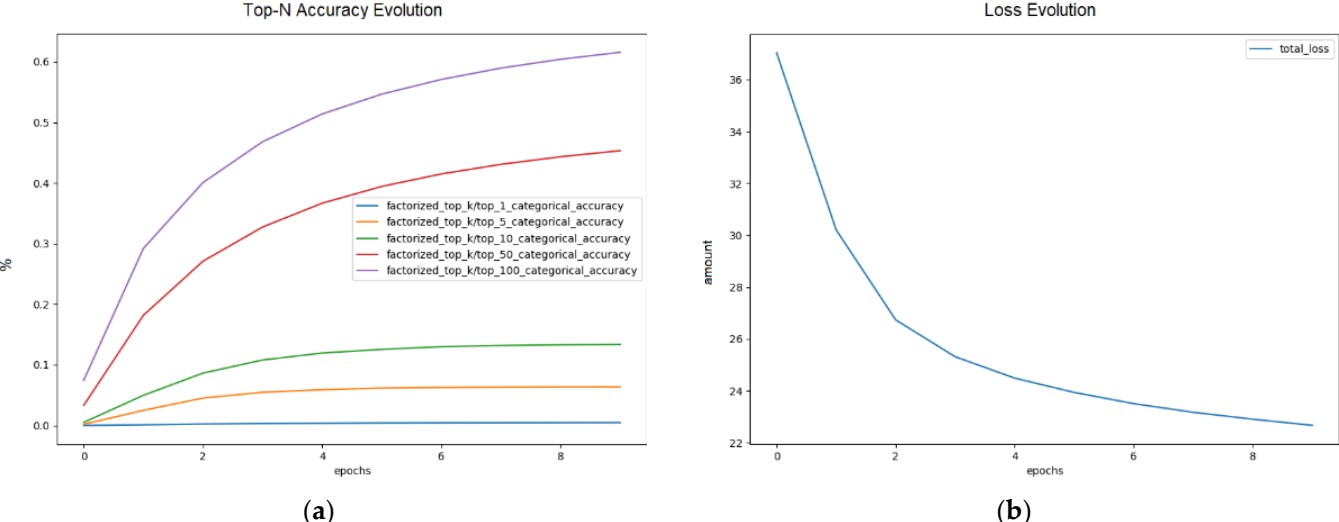

**Figure 8.** Monitoring accuracy (**a**) and loss (**b**) in the training model used with Tensorflow.

The total loss in this second experiment is still considered high for a good model, and under these conditions, recommendations may show few variations in the top-10, top-5 and mainly top-1 among users, as the model has low accuracy. Better scores are obtained for the top-50 and top-100 recommendations. However, the loss falls, and the model used can present even lower values in the total loss and higher accuracy rates. Briefly, hyperparameters can be tuned to ensure a better machine learning setup, such as percentage division between training and testing data, scaling of the characteristics used by embeddings and still learning rate. Wine features such as acidity, body and grape varieties can be added to improve the recommender model.

In these first two experiments focused on the use of the dataset by two different approaches, only user–wine relationships were considered to find relevance in data, and user rating values were not used. Other examples that consider the rating attribute will be presented in Section 3.2, measured by specific metrics.

### 3.2. Study of Evaluation Metrics

The recommendation process is the object of study in many researches and here we only demonstrate a practical use of the X-Wines dataset. However, different models found in the literature can be experimentally executed using different data combinations to be evaluated by known and usual metrics in recommender systems.

The main metrics used are based on data and allow the production of quality scores and their comparison between different heuristics used in the computational models. In traditional machine learning projects, implemented models for classification and regression problems are commonly evaluated using metrics such as accuracy, as well as mean absolute error (MAE) and root mean square error (RMSE) metrics. MAE and RMSE remove the negative sign and calculate the mean error with only the magnitude of the difference found. A difference is that the MAE considers the same weight for any error, while the RMSE squares the error before considering it and tends to penalize the type of error [42].

Specifically in recommender systems, the measurement is made by ranking the average assertiveness among all users, evaluating on a percentage scale whether the desired items are among the top-N recommended items for each user. This way, the smaller the MAE and RMSE values, the better the prediction model tends to be at dealing with outliers. Examples of evaluation metrics commonly used are precision (Precision@N); recall (Recall@N); F1-score (F1@N) harmonic mean between precision and recall; normalized discounted cumulative gain (nDCG@N); hit ratio (Hit@N); mean average precision (MAP); area under the ROC curve (AUC); coverage (COV); average popularity (POP), which nor-

mally do not consider the order of items; and mean reciprocal rank (MRR), which considers the order of items [12,50,53].

In the next experiments, user rating values were considered in addition to user–wine relationships to find relevance in data. Some models use binarization for rated values, considering them as a positive or negative in evaluation. For experimentation purposes, the rating threshold value equal to 4 was defined; that is, all evaluations that received a score between 4 and 5 by the users are considered positive and the other negatives. The casual random split on samples between 80% for training and 20% for testing and the same seed parameter were defined for all. The Cornac framework [54] was used in Python language implementation. This allows comparative experiments on the parametric and multimodal recommender systems. All models were evaluated, considering the average performance for every user being computed first and then the obtained values being averaged to return the final result (executed with the flag *user_based = True*). The default setup from each model presented by their authors was used, including the number of training epochs, which is defined individually. Normally, in recommender systems, the number of training epochs for the model tends to be a small number; that is, few epochs are necessary to obtain a good accuracy fit.

We performed one comparative study of X-Wines dataset against the most referenced MovieLens [2] using some well-known recommender models. MF [14] is a model based on traditional matrix factorization, and weighted matrix factorization (WMF) [55] is an improved over the previous model MF based on calculating a confidence level in the user's positive and negative preference, while singular value decomposition (SVD or SVD++) [56] implements an alternative approach by integrating implicit feedback and transforming both items and users to the same latent factor space. We also employed the KNN user model [51] already used in the first application demo presented in Section 3.1 but now alternating the use between cosine and Pearson similarities. The choice of these models and metrics was merely to be representative of different approaches.

A dataset with the same characteristics of the MovieLens ml_100k version, which contains 100,000 5-stars ratings of 1683 movies rated by 943 users, was eventually selected, with a random selection of the same number of users, wines and ratings among the French wines in X-Wines dataset. In this paper, it is referred to as XWines_100k_FR_Sample. In addition to the default setup used, specifically the same learning rate of 0.001 for models MF and WMF was defined, as well the same number of searched neighbors equal to 10 for KNN models. For the SVD model, the best hyperparameters found in [57] were used, as follows: k = 2 for the number of hidden factors in the factorization, learning rate of 0.0014 and regularization constant equal to 0.08. Tables 2 and 3 present the logs of this simple comparative study.

**Table 2.** Experimental results using MovieLens (ml_100k).

| | AUC | MAP | MRR | nDCG @5 | nDCG @10 | nDCG @100 | Precision @5 | Precision @10 | Precision @100 | Recall @5 | Recall @10 | Recall @100 |
|---|---|---|---|---|---|---|---|---|---|---|---|---|
| **MF** | 0.7254 | 0.0479 | 0.1540 | 0.0558 | 0.0598 | 0.1435 | 0.0534 | 0.0526 | 0.0349 | 0.0191 | 0.0453 | 0.2894 |
| **WMF** | 0.9350 | 0.0675 | 0.1198 | 0.0315 | 0.0421 | 0.2247 | 0.0298 | 0.0374 | 0.0507 | 0.0117 | 0.0424 | 0.5729 |
| **SVD** | 0.7232 | 0.0458 | 0.1406 | 0.0475 | 0.0562 | 0.1322 | 0.0441 | 0.0502 | 0.0322 | 0.0161 | 0.0442 | 0.2640 |
| **UserKNN-Cosine** | 0.6960 | 0.0162 | 0.0150 | 0.0000 | 0.0005 | 0.0374 | 0.0000 | 0.0005 | 0.0118 | 0.0000 | 0.0009 | 0.1025 |
| **UserKNN-Pearson** | 0.6723 | 0.0151 | 0.0143 | 0.0000 | 0.0006 | 0.0280 | 0.0000 | 0.0008 | 0.0098 | 0.0000 | 0.0005 | 0.0711 |

**Table 3.** Experimental results using X-Wines (XWines_100k_FR_Sample).

|  | AUC | MAP | MRR | nDCG @5 | nDCG @10 | nDCG @100 | Precision @5 | Precision @10 | Precision @100 | Recall @5 | Recall @10 | Recall @100 |
|---|---|---|---|---|---|---|---|---|---|---|---|---|
| **MF** | 0.7478 | 0.0488 | 0.1628 | 0.0598 | 0.0583 | 0.1406 | 0.0585 | 0.0534 | 0.0355 | 0.0234 | 0.0397 | 0.2520 |
| **WMF** | 0.8333 | 0.0467 | 0.1261 | 0.0376 | 0.0385 | 0.1445 | 0.0367 | 0.0378 | 0.0406 | 0.0112 | 0.0235 | 0.2868 |
| **SVD** | 0.7545 | 0.0445 | 0.1540 | 0.0508 | 0.0512 | 0.1336 | 0.0478 | 0.0483 | 0.0349 | 0.0172 | 0.0343 | 0.2465 |
| **UserKNN-Cosine** | 0.7714 | 0.0341 | 0.0685 | 0.0150 | 0.0200 | 0.1047 | 0.0180 | 0.0225 | 0.0305 | 0.0072 | 0.0164 | 0.2174 |
| **UserKNN-Pearson** | 0.7620 | 0.0320 | 0.0603 | 0.0111 | 0.0155 | 0.0973 | 0.0138 | 0.0181 | 0.0287 | 0.0048 | 0.0126 | 0.2056 |

The results present in detail the selected metrics in evaluation of the chosen recommender models running training and test processes. The intention was not to compare the indicators obtained by each metric, whether of greater or lesser value, or even to characterize a better or worse result between the datasets. These are different domains (movies and wines), and this comparison type would be guided for algorithms used in the simulation. What can be evaluated in this simulation is that no indicator is zero, even in the top-5 of Table 3, and the resulting values show few variations from other similar indicators in Table 2. The MF and SVD models obtained similar loss of precision and recall in the top-100 between the sample results, different from those other tested models. As a starting point for future research using the X-Wine dataset, the accuracy values are encouraging.

We also exhaustively experimented with the use of data contained in the Slim version presented in Table 1. This was performed to verify the possibility of using this dataset with 12 distinct well-known recommender models and with results being evaluated by various metrics. We again used the Cornac framework to run some different approaches from those used before. The models available in this framework were chosen, being presented with the respective references, on the official page of the Cornac project [58]. The following models were used: probabilistic matrix factorization (PMF), non-negative matrix factorization (NMF), maximum margin matrix factorization (MMMF), Bayesian personalized ranking (BPR), indexable Bayesian personalized ranking (IBPR), item k-nearest-neighbors (ItemKNN), multi-layer perceptron (MLP), neural matrix factorization (NeuMF/NCF), hidden factors and hidden topics (HFT), collaborative topic regression (CTR), variational autoencoder for collaborative filtering (VAECF) and bilateral variational autoencoder for collaborative filtering (BiVAECF).

In this comparative study of models, the default setups presented by the authors were used and indicated by the letter D, and modified hyperparameters, as a first attempt to improve initial results, are indicated by the letter M. Specifically, a data dictionary with up to 5000 most frequent words with the name of the wine, winery and region of the 1007 wines contained in the Slim version was created. This dictionary was used as an input in models above that implement an autoencoder. Model training and testing times in seconds are also shown (a computer with a i7-10750H CPU @ 2.60GHz, NVIDIA GeForce GTX 1650 GPU and 16 GB of RAM on Debian GNU/Linux version 11 was used). The log registry presented in Table 4 remains as a test base for future implementations in this field.

This result presented is promising and can still be improved. In this experimental study, few modifications in some initial hyperparameters such as learning rate, batch size and layers were performed, and better output values were obtained compared to the default hyperparameters in the researched models. The best balance between precision and recall was found in the variational autoencoder for collaborative filtering (VAECF) model [59]. It confirms the possibility of using these data, processed by different algorithms, so that, from the data, new settings and features, it is possible to improve the algorithm scores and offer comparative metric values. Full experimental source-codes and logs accompanying the X-Wines dataset are available in the official repository.

**Table 4.** Experimental results using X-Wines (XWines_Slim_150K_ratings).

| | MAE | RMSE | AUC | F1@10 | MAP | MRR | nDCG@10 | Precision@10 | Recall@10 | Train (s) | Test (s) |
|---|---|---|---|---|---|---|---|---|---|---|---|
| **PMF (D)** [1] | 0.3735 | 0.4218 | 0.8969 | 0.0454 | 0.0681 | 0.0731 | 0.0864 | 0.0259 | 0.2088 | 2.4346 | 2.9242 |
| **PMF (M)** [2] | 0.4100 | 0.4722 | 0.9179 | 0.0550 | 0.1132 | 0.1244 | 0.1340 | 0.0314 | 0.2524 | 4.3186 | 2.8105 |
| **NMF (D)** | 0.3918 | 0.4433 | 0.8631 | 0.0258 | 0.0530 | 0.0588 | 0.0556 | 0.0148 | 0.1178 | 0.4730 | 3.6310 |
| **NMF (M)** | 0.3541 | 0.4025 | 0.8953 | 0.0508 | 0.0903 | 0.0992 | 0.1109 | 0.0290 | 0.2348 | 3.3834 | 3.5685 |
| **MMMF (D)** | 2.8310 | 2.8665 | 0.8426 | 0.0136 | 0.0243 | 0.0273 | 0.0235 | 0.0078 | 0.0636 | 0.1575 | 3.5797 |
| **MMMF (M)** | 2.8310 | 2.8665 | 0.8796 | 0.0311 | 0.0548 | 0.0593 | 0.0639 | 0.0176 | 0.1494 | 0.9520 | 3.6344 |
| **BPR (D)** | 2.2315 | 2.2874 | 0.9026 | 0.0488 | 0.0673 | 0.0742 | 0.0954 | 0.0277 | 0.2329 | 0.2612 | 3.6313 |
| **BPR (M)** | 2.2503 | 2.3100 | 0.9200 | 0.0556 | 0.0956 | 0.1072 | 0.1206 | 0.0317 | 0.2581 | 1.3323 | 3.5726 |
| **IBPR (D)** | 2.8310 | 2.8665 | 0.9030 | 0.0251 | 0.0553 | 0.0620 | 0.0535 | 0.0144 | 0.1137 | 636.7179 | 3.1536 |
| **IBPR (M)** | 2.8310 | 2.8665 | 0.9111 | 0.0362 | 0.0709 | 0.0810 | 0.0789 | 0.0207 | 0.1654 | 896.7034 | 3.3106 |
| **ItemKNN (D)** | 0.4741 | 0.5357 | 0.4759 | 0.0004 | 0.0033 | 0.0038 | 0.0011 | 0.0002 | 0.0019 | 0.2050 | 6.7605 |
| **ItemKNN (M)** | 0.4712 | 0.5341 | 0.6247 | 0.0017 | 0.0078 | 0.0087 | 0.0032 | 0.0010 | 0.0081 | 0.2569 | 7.8847 |
| **MLP (D)** | 2.8310 | 2.8665 | 0.8916 | 0.0514 | 0.1087 | 0.1187 | 0.1269 | 0.0295 | 0.2335 | 182.0409 | 10.1129 |
| **MLP (M)** | 2.8310 | 2.8665 | 0.8975 | 0.0538 | 0.1054 | 0.1155 | 0.1277 | 0.0308 | 0.2470 | 182.7150 | 9.9000 |
| **NeuMF/NCF (D)** | 2.8310 | 2.8665 | 0.8936 | 0.0410 | 0.0850 | 0.0952 | 0.0953 | 0.0236 | 0.1834 | 187.7156 | 11.3031 |
| **NeuMF/NCF (M)** | 2.8310 | 2.8665 | 0.9069 | 0.0461 | 0.0991 | 0.1125 | 0.1122 | 0.0265 | 0.2070 | 229.4710 | 11.4913 |
| **HFT (D)** | 0.7179 | 0.8099 | 0.7332 | 0.0117 | 0.0277 | 0.0314 | 0.0251 | 0.0067 | 0.0525 | 2760.2964 | 3.0833 |
| **HFT (M)** | 0.4068 | 0.4615 | 0.8444 | 0.0326 | 0.0731 | 0.0799 | 0.0799 | 0.0186 | 0.1518 | 181.9239 | 3.1477 |
| **CTR (D)** | 2.7823 | 2.8220 | 0.6216 | 0.0043 | 0.0124 | 0.0139 | 0.0064 | 0.0025 | 0.0181 | 478.4666 | 3.2100 |
| **CTR (M)** | 1.7647 | 1.9052 | 0.8411 | 0.0459 | 0.0975 | 0.1095 | 0.1139 | 0.0262 | 0.2131 | 40.5894 | 3.2452 |
| **VAECF (D)** | 2.8310 | 2.8665 | 0.9224 | 0.0622 | 0.1385 | 0.1553 | 0.1620 | 0.0356 | 0.2843 | 36.1579 | 13.3041 |
| **VAECF (M)** | 2.8310 | 2.8665 | 0.9234 | 0.0646 | 0.1450 | 0.1622 | 0.1700 | 0.0370 | 0.2958 | 39.4341 | 12.5907 |
| **BiVAECF (D)** | 2.7008 | 2.7369 | 0.9145 | 0.0450 | 0.0788 | 0.0871 | 0.0970 | 0.0257 | 0.2090 | 68.9236 | 4.3243 |
| **BiVAECF (M)** | 2.5775 | 2.6160 | 0.9212 | 0.0631 | 0.1237 | 0.1386 | 0.1526 | 0.0360 | 0.2915 | 164.4968 | 4.4348 |

[1] (D): Default hyperparameters. [2] (M): Modified hyperparameters

## 4. Conclusions

This paper introduces and characterizes X-Wines, a wine dataset built during six months of pre-processing, data validation and verification. It aims to offer a large and consistent data volume to the scientific community. X-Wines is openly available to support educational, research and general-purpose projects, especially those that require a large quantity of data.

We removed conflicts of interest that may exist in data when made available under a free license. Our contribution lies in the pre-processing work and in confirmation of use because limitations were found in information and formats made available on the open Web. Not all producers systematically disclose their products with all the relevant information. Many attributes are not openly found on the Web and required several checks to be validated. Vinification processes may vary by vintages, and the choice of attributes presented in this dataset to the detriment of other candidates was limited by the quantity and the quality of information found.

Recommender systems currently used, including deep learning models, are being run and evaluated offline. The presented results show that the X-Wines dataset is perfectly applicable in recommender system algorithms. The evaluation by specific metrics, as well as their models created by different approaches, will continue to be improved through further research, and they will need data to evolve. X-Wines stands out as a new reference dataset for wider free use.

Soon, data updates to new versions of the X-Wines dataset will aim to provide even more quality information. We are working to keep the dataset in line with FAIR principles and hope to include implicit user feedback from logging sessions through a collaborative platform. We have interest in finding realistic recommender system practices meeting user needs to mitigate information overload in online environments, especially considering short-term and long-term sequential dynamics.

**Author Contributions:** Conceptualization and analysis, R.X.d.A., A.J.M. and V.F.; methodology, software, data collection, validation, R.X.d.A.; writing—original draft preparation, R.X.d.A.; writing—review and editing, R.X.d.A., A.J.M. and V.F.; supervision, A.J.M. and V.F. All authors have read and agreed to the published version of the manuscript.

**Funding:** This research received no external funding.

**Institutional Review Board Statement:** Not applicable.

**Informed Consent Statement:** Not applicable.

**Data Availability Statement:** A publicly available dataset was built in this study and others publicly datasets were analyzed. These data and the source codes of the experiments carried out are available online at https://github.com/rogerioxavier/X-Wines (accessed on 26 December 2022).

**Acknowledgments:** This research has been supported by the Instituto Federal de Educação, Ciência e Tecnologia do Rio Grande do Sul (IFRS)—Brazil.

**Conflicts of Interest:** The authors declare no conflict of interest.

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
