# Peer review of "X-Wines: A Wine Dataset for Recommender Systems and Machine Learning"

_2504-2289, doi:10.3390/bdcc7010020_

Round 1

Reviewer 1 Report (Previous Reviewer 3)

I would like to thank the authors for addressing the comments in my previous review.

Author Response

Dear Reviewer,

We thank you for your contributions, they were very important for us to be able to improve our work.

We hope you can make use of the X-Wines dataset if you wish or even encourage its use.

Thank you very much.

The authors.

Reviewer 2 Report (Previous Reviewer 1)

I thank the authors for addressing the issues that I raised in the previous version. The paper improved substantially. However, there are still several major issues with the current version of the paper. In particular:

- English needs to be significantly improved. There are multiple grammatical/stylistic errors. Because of that it is extremely difficult to follow the paper. Frequently, I was also not able to understand the meaning of the sentences.

- Some of the decision/claims are not typical. For example, in Section 2.3. the authors state "A manual statistical test was conducted to assess the reality of the data obtained from the Web". I can not understand what is meant by that. The data is collected from the Web and it is a Web sample and standard inferential methods may be used to estimate statistics of interest and their confidence intervals. I am not aware of any tests that assess the "reality" of data.

- Similarly, in Section 4.1. I did not understand the experimental setup. What was the training/test split. How many times? Has data been split randomly? Why is the evaluation performed only on a single user? How was this user selected?

Author Response

Dear Reviewer,

We thank you for your contributions, they were very important for us to be able to improve our work. We present reviewer transcriptions in blue and authors’ answers in green.

“English needs to be significantly improved. There are multiple grammatical/stylistic errors. Because of that it is extremely difficult to follow the paper. Frequently, I was also not able to understand the meaning of the sentences.”

  1. We officially inform you that the submitted article has been completely reconstructed based on the considerations of the reviewers. The article was revised and corrected by a native speaker from London-UK at CES-Centre of English Studies. A textual reorganization and new verification of technical aspects were carried out in the language, to clarify possible doubts and difficulties of readers that are not experts in the scientific area of the paper. We are also willing to specifically present any specific clarification on the text that may not yet have been identified. Thank you very much.

“Some of the decision/claims are not typical. For example, in Section 2.3. the authors state "A manual statistical test was conducted to assess the reality of the data obtained from the Web". I can not understand what is meant by that. The data is collected from the Web and it is a Web sample and standard inferential methods may be used to estimate statistics of interest and their confidence intervals. I am not aware of any tests that assess the "reality" of data.”

  1. Section 2.3 (now 2.4) was added to the article by request of another reviewer, who suggested us on how to proceed in carrying out a statistical test. They suggested a manual check on a sample of 100 wine items randomly selected from the dataset. We agreed that it would be possible, and it could improve the data trust obtained from the open web and validated by electronic procedures. The reviewer was completely satisfied, approving the article with ‘Yes’ in all their evaluations. Likewise, we consider your suggestion very important and have rewritten the statement in order to remove doubts in this regard. Thus, “… a statistical test was conducted to estimate confidence of the data obtained from the Web…” and adjusted in other points related to this. In addition, other statistical measures were presented, such as standard deviation and confidence interval of the result found. We understand that this was a great contribution to qualify even more the work done. Thank you very much.

“Similarly, in Section 4.1. I did not understand the experimental setup. What was the training/test split. How many times? Has data been split randomly? Why is the evaluation performed only on a single user? How was this user selected?”

  1. Previously you wrote: “...including source code in the main text of the paper does not add scientific value to the paper and should be removed. The source code can be published in an online repository and a link should be included in the paper.”  We considered this important and removed that part, however we also considered that it could cause some lack of information about some of the algorithms used, even though all the information may be available in the repository indicated in the text. We revised section 4.1 (now 3.1) and added some information in order to better characterize the experiments carried out. It is important to notice that the focus of the article is on the dataset and its usefulness, and the experiments were added as simple examples of application. We believe that the result has been quite didactic for the readers.

Finally, we advanced to qualify all parts of the text in this latest version with more information, more details of the results expressed in images, graphs and tables, including a comparison with the most quoted dataset in the recommender systems literature (Tables 2 and 3). We ran X-Wines on over 15 different classic algorithms and presented the results (Table 4). We followed many of your contributions, which we really consider very important to improve the article’s quality. Even not having found any notes for improvement in the introduction and presentation of results, for example, we also worked on those sections, as X-Wines dataset will certainly be a good contribution to the scientific community. We hope you can make use of the X-Wines dataset if you wish or even encourage its use through this article.

Thank you very much.

The authors.

We provide complementary material in the X-Wines project repository at https://github.com/rogerioxavier/X-Wines

Reviewer 3 Report (Previous Reviewer 2)

Author Response

Dear Reviewer,

We thank you for your contributions, they were very important for us to be able to improve our work. We officially inform you that the submitted article has been completely reconstructed based on the considerations of the reviewers and one reviewer is completely satisfied. We advance to qualify all parts of the text in this latest version with more information, greater detailing of the results expressed in images, graphs and tables. We run X-Wines on over 15 different classic algorithms and present the results from various evaluation metrics (Table 4). We follow many of your contributions, which we really consider very important to improve the article’s quality. We present reviewer transcriptions in blue and authors’ answers in green.

Previously you wrote “1. A proper literature review is missing. Currently, the paper has just 2 sentences and 4 references as far as the review of prior works is concerned, which is insufficient for a journal paper.”

Our first answer: “ 1. Yes, we have identified this gap and we want to start filling it. We believe have better explained this lack with a better literature review. We also considered it necessary and now we had more time to do it.“

Now, there are no comments.

  1. We believe we have overcome this point; however, we are constantly reviewing it to reach the final version.

2. “I understand the point. However, it is recommended that a statement(s) from the privacy policy and/or content re-distribution policy from these websites are quoted to support the fact that the specific policies allow mining the specific attributes of the data that the authors mined from these websites.

For instance, a slightly unrelated example:

The guidelines for Twitter content redistribution state “If you provide Twitter Content to third parties, including downloadable datasets or via an API, you may only distribute Tweet IDs, Direct Message IDs, and/or User IDs”.

Link for Twitter’s policy on content redistribution: https://developer.twitter.com/en/developer-terms/agreement-and-policy

For this research paper, Twitter’s policy is not applicable but it is quoted here for a clarification for the authors on the kind of facts/statements that may be presented from specific and applicable policies to address this concern.”

  1. We have obtained the agreement of portals’ owners to use the public data available in their websites. Wineries are also glad to advertise their products. This scenario is perfectly fair and acceptable by all parts. There was no objection and the decision not to identify the users was ours, even before owners requested that (some did not).

We collect only the necessary data from specific dates. We understand that new collections will require new permissions. We believe that the most important issue was presented in the text: the world dataset produced is an unpublished Web sample, without identifying people, and will be distributed under a free license for its wider use. We are still providing an ODC-by or ODC-ODbL license from Open Data Commons (opendatacommons.org), or even a different one, we need to evaluate which will present a better definition when it comes to a dataset. Thank you very much.

3.“Multiple references have been cited in the Introduction section. However, several references that have been cited are very old. For instance, [9] was published 17 years ago, [8] was published 13 years ago and so on. Consider updating such old references by the two papers that were suggested in the last review round (see comment above).”

  1. We understand the point. We always prefer to use the most recent references, however in this specific case, both [8] and [9], were being used in a historical context. In the introduction, we talk first about wines, then about the large amount of data presented on the Web and cite some of the best-known datasets; then at this mentioned point, we start talking about recommender systems, who first defined and classified these systems, in a historical context, until we identified today the lack of a dataset like X-Wines, which will certainly be very useful to the scientific community. Mentioned reference [9] [Adomavicius & Tuzhilin, 2005] is very important in the historical context of recommender systems (Table 5 in https://doi.org/10.1016/j.eswa.2020.113764). In this case, we prefer to go directly to the source than consider other references that may also be important.

However, we considered it a good suggestion in some cases, and we added more recent references about filtering techniques using artificial intelligence in recommender systems. We can remove [8] and cite reference [9] (now [41]) in chapter 3 by highlighting its real contribution in the recommender systems definition.

“4. This statement has been added to the paper – ‘The developed processes are in line with FAIR — Findable, Accessible, Interoperable and Reusable guiding principles’ but the clarification is missing. The response provided by the authors here is not relevant. The authors state – ‘the FAIR principles support the data management needs of scientists and researchers’. There wasn’t a question about what FAIR principles support. The concern here is that it is not clear whether the presented dataset is in compliance with the FAIR principles. Please explain this with proper clarifications. Specifically, what properties of this dataset make it Findable? What properties make it Accessible? What properties make it Interoperable? And what properties make it Reusable?”

  1. Clearly the FAIR principles bring substantial support to a work like this. Considering this contribution very important, we decided to make the proper reference, since it showed great compatibility with the work carried out. Thank you very much, as it was considered a gain and a beautiful contribution and now its presence in the text is unquestionable. However, we need to deal with the form of presentation. We would like to inform you that our work has not been guided by these principles as a guideline. When we started in the first half of 2022, we did not have this commitment. We followed the work methodology presented in the article and produced an unpublished dataset. Our work was guided by the principles of science and ethics. Even if not explicitly cited in the text, as perhaps you would like to find everything in a single paragraph, the principles are presented throughout the text body, without a checklist, but including with more than one citation about it. When we talk about data extraction protocol only in authorized platforms, publication of all source URLs for proper verification of origin, annotation of important information and processes by machine actionable, among others. By allowing the data sequences construction we make it possible to reuse data in recommendation systems, for example. If the presence of another more specific text on the subject is considered majority, we will be able to build smoothly. This could modify the current textual structure that started without FAIR; however, a specific paragraph was added, but without deviating from the work carried out and modifying the results presented. We understand that this was a great contribution to qualify even more the work done.

Previously you wrote “5. Why was kNN used as the approach to develop the recommendation system? Why were other classification approaches, such as random forest or gradient-boosted trees, not used?”

Our first answer: “ 5. Among other possibilities found in the reference literature, two usual applications were chosen to demonstrate the applicability of the dataset in recommender systems. We do not want to make the paper too long, as the focus is on disseminating the dataset to the scientific community. We review and rearranged the chapter and improve the results presented. Various models built by different approaches have now been used.”

Now, there are no comments, but it was found “The authors state – ‘we don't want to extend the article too much’. In response to comment #5, the authors stated – ‘We do not want to make the paper too long’. Please note that the journal website (https://www.mdpi.com/journal/BDCC/instructions) states – ‘BDCC has no restrictions on the length of manuscripts’

  1. Like point 1, we believe we have overcome this point; however, we are constantly reviewing it to reach the final version. The reference to the number of pages is because in this case we consider that the extent of the article is adequate to this theme. If the focus was on novel applications or algorithms, we would consider more pages. Thank you very much.

6. “Tables 2 and 3 certainly present some results that were not presented before. Please compare these results with any similar works in this field to highlight the significance and relevance of these findings. A comparative study (using these results) with prior works is also suggested to highlight how these findings outperform the findings of prior works in this field.”

  1. Thank you for acknowledging progress in the inclusion of section 4.2 (now 3.2). This was your suggestion and we have developed it. We include a use comparison of X-Wines with the most quoted dataset MovieLens in the recommender systems literature (tables 2 and 3). We use classical algorithms well-known in the reference literature, various classification approaches and evaluation metrics. Now, we reorganized section 3.2 and added some information in order to better characterize the experiments carried out. The first study was redone using best hyper-parameters for SVD model. Even if the focus of the article is on the dataset and its usefulness, certainly many other algorithms will use this dataset in the future with distinct emphases considering the most varied setups. We accept your suggestion and intend to do this in the future when the focus of our research is on the performance of new proposed algorithms. New comparisons deserve to be presented in specific articles, because they could take months to develop (for example, the table 3 in https://doi.org/10.3390/app10144926). In this article we introduce X-Wines to the community and instigate many other works that can use it widely. We considered that other comparisons, as well as, the source code can be published in the online repository, including gathering the best works of other authors who used X-Wines.

Previously you wrote: “7. The title says ‘software applications’ but the paper doesn't provide a lot of information on the specific kind of software applications that may find this dataset useful.”

Our first answer: “The new dataset will be openly published for wider free use. It will certainly be referenced in several areas of knowledge and by different software applications, although some may be explicitly mentioned in the text, expressing their own characteristics.”

Now, there are no comments.

  1. As previously mentioned, we decided not to limit the dataset use, if some areas are described in the text body, and to remove the questioned part of the title (“… and other applications”) for this new article. We believe we have overcome this point. Thank you very much.

Finally, even not having found any notes for improvement in several chapters and having found some improvements recognized, we are always reviewing anything that might improve this paper, as X-Wines dataset will certainly be a good contribution to the scientific community. We hope you can make use of the X-Wines dataset if you wish or even encourage its use through this article.

Thank you very much.

The authors.

We provide complementary material in the X-Wines project repository at https://github.com/rogerioxavier/X-Wines

Round 2

Reviewer 3 Report (Previous Reviewer 2)

The authors have revised their paper as per my comments and suggestions. I do not have any additional comments at this point. I recommend the publication of the paper in its current form. 

This manuscript is a resubmission of an earlier submission. The following is a list of the peer review reports and author responses from that submission.

Round 1

Reviewer 1 Report

The paper presents a new wine dataset that can be used for building wine recommender systems. While the authors correctly recognize the lack of useful large-scale datasets with wine ratings, the overall scientific significance and the impact of the paper must be improved before the paper can be published. For example, the demonstration of the use-cases in the second part of the paper could be extended with more methods for recommendations including some of the current state-of-the-art methods. The methods need to be evaluated by a standard recommender metrics such as precsion, recall, or NDCG. Such an evaluation would clearly demostrate the utility and usefulness of the dataset and would provide a base for further implementations in this area.

The data collection itself is described, but the authors need to provide further justification for the collection and preprocessing steps that they took.

Also, the presentation of the paper and English need to be significantly improved. For example, including source code in the main text of the paper does not add scientific value to the paper and should be removed. The source code can be published in an online repository and a link should be included in the paper.

Author Response

Dear Reviewer,

We appreciate all your considerations. They were very welcome and very important in the reconstruction of this article. We are now sending a new manuscript with greater detail, because we had more time to make the suggested adjustments, we also added more information and results. The dataset is very important. We believe it will be useful for the scientific community. We have chosen this article to be cited by anyone who uses X-Wines. It will certainly be referenced a lot in the future.

So we answer:

“The paper presents a new wine dataset that can be used for building wine recommender systems. While the authors correctly recognize the lack of useful large-scale datasets with wine ratings, the overall scientific significance and the impact of the paper must be improved before the paper can be published. For example, the demonstration of the use-cases in the second part of the paper could be extended with more methods for recommendations including some of the current state-of-the-art methods. The methods need to be evaluated by a standard recommender metrics such as precsion, recall, or NDCG. Such an evaluation would clearly demostrate the utility and usefulness of the dataset and would provide a base for further implementations in this area.”

We believe this has been fully resolved. This work focused on the presentation this new dataset to the scientific community, which will be able to wider free use. A better review on the topic was built and referenced. We improve the experiments, reducing the source codes previously presented, using other algorithms and presenting quantitative results such as the metrics indicated and others metrics, that we are used to, in graphs, tables and senteces.

“The data collection itself is described, but the authors need to provide further justification for the collection and preprocessing steps that they took.
Also, the presentation of the paper and English need to be significantly improved. For example, including source code in the main text of the paper does not add scientific value to the paper and should be removed. The source code can be published in an online repository and a link should be included in the paper.”

Perfectly, this has been fully realized. We performed a detailed review of the article's writing. It is practically a new article. Introducing greater detail in data collection and results.

We really appreciate your contributions. Thank you very much.

The authors

PS. We provide complementary material in the X-Wines project repository at https://github.com/rogerioxavier/X-Wines

Reviewer 2 Report

This paper is titled – “X-Wines: A wine dataset for recommender systems, machine learning, and other software applications”. This work presents X-Wines, a new wine dataset containing 21 million real evaluations from registered users on specialized wine platforms. Data were collected between February and March 2022 by data-scrape processes on the open Web. Classic recommender systems with deep learning algorithms were run employing X-Wines as a specific use case. The dataset seems to have value for different applications. However, parts of the paper need major improvement. It is suggested that the authors make the necessary changes/updates to their paper as per the following comments:

1. A proper literature review is missing. Currently, the paper has just 2 sentences and 4 references as far as the review of prior works is concerned, which is insufficient for a journal paper. 

2. The authors performed web scraping on different websites to mine the data. Is web scraping allowed by these respective websites? Does using a web scraper to obtain and re-use the data (to develop a dataset) follow the protocols of terms and conditions and content redistribution of these websites? Please provide relevant references pointing to these terms and conditions, content redistribution policies, etc., while clarifying these concerns.

3. The introduction section is very brief and written like a conference paper. The authors should clearly highlight the relevance of datasets in general and cite a few recent papers that focused on the development of datasets to support the discussion. Consider citing these 2 recent papers on datasets  https://doi.org/10.3390/data6080092 and https://doi.org/10.3390/data6080090, which have attracted multiple citations (especially the first one)

4. The FAIR principles for scientific data management [reference: https://www.nature.com/articles/sdata201618] state that a dataset should have Findability, Accessibility, Interoperability, and Reusability. The paper does not mention anything about compliance with FAIR principles.

5. Why was kNN used as the approach to develop the recommendation system? Why were other classification approaches, such as random forest or gradient-boosted trees, not used?

6. Please provide specific metrics to justify the accuracy of the recommendation system that was built. 

7. The title says "software applications" but the paper doesn't provide a lot of information on the specific kind of software applications that may find this dataset useful. 

Author Response

Dear Reviewer,

We appreciate all your considerations. They were very welcome and very important in the reconstruction of this article. We are now sending a new manuscript with greater detail, because we had more time to make the suggested adjustments, we also added more information and results. The dataset is very important. We believe it will be useful for the scientific community. We have chosen this article to be cited by anyone who uses X-Wines. It will certainly be referenced a lot in the future.

So we answer:

1. Yes, we have identified this gap and we want to start filling it. We believe have better explained this lack with a better literature review. We also considered it necessary and now we had more time to do it.

2. This is a concern we have as researchers. We respect data protection laws and data ownership. We contacted specialized wine portals and wineries. We explained the work that was being done and there was no opposition. It was considered very interesting. Wine producers and traders need to promote their products and we do not work with private data, as explained below. As return, we received information that the portals do not want to send their data, but what is presented on the open Web could be used to complete our registrations. We only need to identify the wines but not the users. We have authorization,  however it is very difficult to obtain formal authorization to perform the practice on particular websites. Which is quite understandable.
Wine data is present and replicated on a large scale on the Web. Many people understand that Web scraping is not an illegal practice, it is a technological resource and can be put to good use. Companies like Google use scraping to work their indexes. However, the term scraping was removed from the text so as not to sound pejorative. Besides meeting no objection, some contacts were grateful us for exposing their products and trademarks.
Data was collected from public Web pages, where users did share some personal data. However, users were not explicitly identified in X-Wines, preserving their identity. Only a sequential user ID is available in X-Wine to identify a specific anonymous user. This is the procedure required by law both in Brazil and Portugal. 
Only data sheets made available by wineries and trades were used, as well as data openly available on platforms specialized in wines. These platforms work as social networks, forming clubs and communities interested in a particular product. As found, the specialized wine platforms ask their clients whether or not they want to publish their wine reviews to help other people.

3. The introduction section, like other parts of the text, was improved and is already revised. We also change some figures, we removed source codes available in our repository and mainly reorganized the second part of the article. Thank you for indicating the references, we are already analyzing this material for use in the future.

4. Perfectly, the FAIR principles support the data management needs of scientists and researchers.

5. Among other possibilities found in the reference literature, two usual applications were chosen to demonstrate the applicability of the dataset in recommender systems. We do not want to make the paper too long, as the focus is on disseminating the dataset to the scientific community. We review and rearranged the chapter and improve the results presented. Various models built by different approaches have now been used.

6. Yes, this needs to be presented. We improve the experiments, reducing the source codes previously presented and which are in the official repository, detailing other models and presenting quantitative results such as the metrics precsion, recall, NDCG, MAP, AUC and others that we are used to, in graphs, tables and senteces.

7. The new dataset will be openly published for wider free use. It will certainly be referenced in several areas of knowledge and by different software applications, although some may be explicitly mentioned in the text, expressing their own characteristics. However, we decided not to limit its use and remove this part of the title. The most important is to present and characterize the dataset and we don't want to extend the article too much, as we already have 15 pages. So we focus on results.

We really appreciate your contributions. Thank you very much.

The authors

We provide complementary material in the X-Wines project repository at https://github.com/rogerioxavier/X-Wines

Reviewer 3 Report

The paper focuses on the creation of a wine dataset containing 21 million evaluations. The data has been obtained through web scraping.

The topic is interesting and worth investigating. The dataset is sufficiently well described, and the paper also showcases how it can be used.

The paper should better describe how the authors have made sure that the information extracted is correct. For example, the authors could choose to manually analyze the information for 100 wines and see for each attribute the percentage of correct values.
Additionally, the paper mentions that 350 sources have been found for extracting information related to wines. The full list of websites that have been crawled should be made available either on GitHub or as a supplementary material to the paper.
The paper should also clarify the meaning of “process up to 10 levels deep into Web browsing”.
The authors are kindly asked to better explain the parsing process. Has specific code been written for each of the 350 identified information sources.  
The paper mentions that some data has been extracted from CSS. CSS is a language normally used for formatting (ex: such as making a text red). In this context, the authors are kindly asked to provide a few examples of data that has been extracted from CSS, also mentioning the URL of the corresponding CSS.
The paper should also better explain how data has been extracted from pdf, txt and epub files and how the authors have made sure that the extracted information is correct

Author Response

Dear Reviewer,

We appreciate all your considerations. They were very welcome and very important in the reconstruction of this article. We are now sending a new manuscript with greater detail, because we had more time to make the suggested adjustments, we also added more information and results. The dataset is very important. We believe it will be useful for the scientific community. We have chosen this article to be cited by anyone who uses X-Wines. It will certainly be referenced a lot in the future.

So we answer:

“The paper focuses on the creation of a wine dataset containing 21 million evaluations. The data has been obtained through web scraping.
The topic is interesting and worth investigating. The dataset is sufficiently well described, and the paper also showcases how it can be used.”

Much of the machine learning processing work is done with data preparation. It can be hard work. This research aims contribute in the wine domain and support future works. We detail the processes carried out and add results that make useful the presented and described dataset.

“The paper should better describe how the authors have made sure that the information extracted is correct. For example, the authors could choose to manually analyze the information for 100 wines and see for each attribute the percentage of correct values.”

Yes, that had been done and now it was presented. Electronic verification was used for such an expressive number of validated data. Several algorithms were used, some in very specific cases. Candidate wine instances and rating objects were created with their respective attributes. Each data attribute had several other control attributes, usually Boolean, which were updated with each new load of extracted data. Algorithms with Python SequenceMatcher functions and Cosine similarity were used to find matches between the raw data. 
A statistical evaluation comparing data from the dataset against real data has already been performed manually and is now presented. The results were very good.

“Additionally, the paper mentions that 350 sources have been found for extracting information related to wines. The full list of websites that have been crawled should be made available either on GitHub or as a supplementary material to the paper.”

Yes we do. We will be able to publish a list with more than 15,000 URLs used in data extraction, mostly producer websites. Even as contribution to the dissemination of their products and trademarks.

“The paper should also clarify the meaning of “process up to 10 levels deep into Web browsing.
The authors are kindly asked to better explain the parsing process. Has specific code been written for each of the 350 identified information sources.” 

Searched source URLs always indicate new links to other sites. For example, when an html <href> tag is detected, the algorithms recursively checked its new content, limited to 10 URLs in a sequential line (levels). This data stacking was explained in the paper.

“The paper mentions that some data has been extracted from CSS. CSS is a language normally used for formatting (ex: such as making a text red). In this context, the authors are kindly asked to provide a few examples of data that has been extracted from CSS, also mentioning the URL of the corresponding CSS.”

Please, CSS format was an error. Thank you very match! The correct one is XML, being used data texts in HTML, Json and XML formats. Even though minority data in XML format were also collected.

“The paper should also better explain how data has been extracted from pdf, txt and epub files and how the authors have made sure that the extracted information is correct”

Data collection in pdf, txt (majoritary) and epub (minority) files used the OCR functions in the Python language, which in turn, use data dictionaries from wine domain for electronic validation, for example countries, grapes and others. When searching files, the frequency and type of data are considered neighboring key-words, such words as: ABV, alcohol, acidity, PH, body, grape, ... The clarification is timely, but we also thought not to extend the article too much, as we already have 15 pages. So we focus on results.

We really appreciate your contributions. Thank you very much.

The authors

We provide complementary material in the X-Wines project repository at https://github.com/rogerioxavier/X-Wines